# Dpb4 promotes resection of DNA double-strand breaks and checkpoint activation by acting in two different protein complexes

Erika Casari [1], Elisa Gobbini[1], Marco Gnugnoli [1], Marco Mangiagalli [1], Michela Clerici [1] & Maria Pia Longhese [1]✉

Budding yeast Dpb4 (POLE3/CHRAC17 in mammals) is a highly conserved histone fold protein that is shared by two protein complexes: the chromatin remodeler ISW2/hCHRAC and the DNA polymerase ε (Pol ε) holoenzyme. In *Saccharomyces cerevisiae*, Dpb4 forms histone-like dimers with Dls1 in the ISW2 complex and with Dpb3 in the Pol ε complex. Here, we show that Dpb4 plays two functions in sensing and processing DNA double-strand breaks (DSBs). Dpb4 promotes histone removal and DSB resection by interacting with Dls1 to facilitate the association of the Isw2 ATPase to DSBs. Furthermore, it promotes checkpoint activation by interacting with Dpb3 to facilitate the association of the checkpoint protein Rad9 to DSBs. Persistence of both Isw2 and Rad9 at DSBs is enhanced by the A62S mutation that is located in the Dpb4 histone fold domain and increases Dpb4 association at DSBs. Thus, Dpb4 exerts two distinct functions at DSBs depending on its interactors.

[1] Dipartimento di Biotecnologie e Bioscienze, Università degli Studi di Milano-Bicocca, Milano, Italy. ✉email: mariapia.longhese@unimib.it

DNA double-strand breaks (DSBs) are harmful genomic lesions that threaten genome stability and cell survival. Eukaryotic cells use two main pathways for the repair of DSBs: non-homologous end-joining (NHEJ) and homologous recombination (HR)[1,2]. HR requires that the 5′ strands at both DSB ends undergo nucleolytic degradation (resection), generating 3′-ended single-stranded DNA (ssDNA) tails that can invade the undamaged homologous DNA template[3]. DSB resection is initiated by the binding to the DSB ends of the evolutionarily conserved Mre11-Rad50-Xrs2/NBS1 (MRX/N) complex[4]. The Sae2 protein (CtIP in mammals) activates a latent endonuclease activity of Mre11, which cleaves the 5′-terminated strands at both DNA ends[5]. The resulting nick generates an entry site for both Mre11, which degrades back toward the DSB end in a 3′–5′ direction, and the long-range resection Exo1 and Dna2 nucleases, which catalyze extended resection in a 5′–3′ direction away from the DSB[6–13]. The MRX complex is also necessary to recruit Exo1 and Dna2 to DSBs[3].

Repair of DNA DSBs occurs within chromatin, raising the question of how DSBs can be detected, signaled, and repaired within this context. Different subfamilies of chromatin-remodeling enzymes catalyze a broad range of chromatin modifications, which include sliding histone octamer across the DNA, changing the conformation of nucleosomal DNA, or the composition of the histone octamer[14]. Eukaryotes have four subfamilies of chromatin-remodeling factors, namely SWI/SNF, ISWI, CHD, and INO80/SWR. In yeast, the RSC and the SWI/SNF complexes, two members of the SWI/SNF chromatin remodeler family, promote MRX recruitment to DSBs and DSB resection by catalyzing eviction of nucleosomes adjacent to a DSB[15–17]. ssDNA generation at the DSB ends requires also the Ino80 complex that participates in the eviction of nucleosomes on either side of a DSB[18–21].

Generation of DSBs elicits a cellular response, termed DNA damage checkpoint, that senses DNA damage and transduces this information to regulate several cellular processes, including cell cycle progression, DNA repair, and DNA replication[22]. Key players of the checkpoint cascade include the *Saccharomyces cerevisiae* protein kinases Mec1 and Tel1, as well as their mammalian orthologs ATR and ATM[23]. Upon DNA damage recognition, these apical kinases activate the downstream effector kinases Rad53 (CHK2 in mammals) and Chk1. Rad53 and Chk1 activation require Rad9, which acts both as an adaptor between Mec1 and Rad53 and as a scaffold to promote Rad53 autophosphorylation and activation[24–26].

In *S. cerevisiae*, Mec1 activation requires additional factors including the highly conserved Ddc1-Mec3-Rad17 (hereafter called 9-1-1) complex and the replication factor Dpb11 (TopBP1 in mammals)[27–29]. The 9-1-1 complex, structurally related to the replication sliding clamp PCNA, recruits to DNA damaged sites Dpb11, which in turn interacts with the checkpoint protein Rad9[30–32]. Dpb11–Rad9 interaction requires cyclin-dependent kinase (Cdk1)-mediated Rad9 phosphorylation on the S462 and T474 residues, which bind directly to the N-terminal domain of Dpb11[32].

Dpb11 is also part of the DNA polymerase ε (Pol ε) holoenzyme, which is largely responsible for leading-strand synthesis during DNA replication. Pol ε consists of Pol2, Dpb2, Dpb3 (POLE4 in mammals), and Dpb4 (POLE3/CHRAC17 in mammals) subunits[33–35]. Both Dpb3 and Dpb4 contain a histone fold domain, through which they interact to form a H2A–H2B-like complex that is not essential for cell viability in budding yeast[36,37]. In both yeast and mammals, the Dpb3–Dpb4 complex binds H3–H4 tetramers and facilitates their transfer onto the leading strand during DNA replication through an intrinsic chaperone activity[38,39]. Interestingly, genetic studies reveal a role

for Dpb3 and Dpb4 in maintaining the silenced state of chromatin[40–42], suggesting that a defect in parental H3–H4 transfer in *dpb3Δ* and *dpb4Δ* cells might compromise epigenetic inheritance. The maintenance of heterochromatin silencing also involves the catalytic subunit of Pol ε[40,41] and this function appears to be dependent upon Dpb3 and Dpb4, which bind double-stranded DNA (dsDNA) and increase Pol ε association to it[43].

Of note, in yeast, *Drosophila melanogaster*, and humans, Dpb4/POLE3/CHRAC17 is also a component of the ISW2/hCHRAC chromatin-remodeling complex[40,44,45], which catalyzes nucleosome sliding through the ATPase motor protein Isw2/hSNF2H[46]. In the budding yeast ISW2 complex, Dpb4 interacts with the histone fold protein Dls1 that is considered a Dpb3 paralog[47]. In the mammalian ISW2 complex, the catalytic hSNF2H subunit has been implicated in the repair of DSBs by stimulating the association to them of the recombination protein BRCA1[48], while the noncatalytic ACF1 subunit directly interacts with the NHEJ protein complex KU70-KU80 and promotes its accumulation to DSBs[49,50].

Here we show that the lack of *S. cerevisiae* Dpb4 reduces both histone removal from the DSB ends and MRX accumulation at DSBs. The poor MRX retention in *dpb4Δ* cells leads to a defective DSB resection. Furthermore, the lack of Dpb4 impairs activation of the checkpoint response by reducing Rad9 association to DSBs. Dpb4 promotes DSB resection and checkpoint activation by acting in two different protein complexes. In fact, Dpb4 interacts with Dls1 to promote Isw2 association to DSBs, histone removal, and DSB resection, while it interacts with Dpb3 to promote Rad9 association to DSBs and checkpoint activation.

## Results

**The *dpb4-A62S* allele exacerbates the sensitivity to camptothecin of *tel1Δ* and *sae2Δ* cells more severely than *DPB4* deletion.** Cells lacking Tel1 are specifically sensitive to camptothecin (CPT)[51], which stabilizes DNA topoisomerase I cleavage complexes, yielding to replication-dependent DSBs[52]. We have previously searched for extragenic mutations that exacerbated the CPT hypersensitivity of *tel1Δ* cells[53]. Genome sequencing and genetic analysis revealed that one of the mutations responsible for the increased CPT sensitivity of *tel1Δ* cells was a single nucleotide change in the *DPB4* gene that caused the replacement of Ala62 with Ser. The synthetic cytotoxicity caused by the *dpb4-A62S* allele turned out to be not specific for *tel1Δ* cells (Fig. 1a), as the same mutation also exacerbated the sensitivity to CPT of *sae2Δ* cells (Fig. 1b).

To understand whether the *dpb4-A62S* mutation exacerbates the DNA damage sensitivity of *tel1Δ* and *sae2Δ* cells by disrupting Dpb4 function, we analyzed the effect of *DPB4* deletion. *dpb4Δ tel1Δ* and *dpb4Δ sae2Δ* cells were less sensitive to CPT than *dpb4-A62S tel1Δ* and *dpb4-A62S sae2Δ* cells, respectively (Fig. 1a, b), suggesting that the synthetic effect caused by the *dpb4-A62S* allele is not due to loss of Dpb4 function. Although the *DPB4* deletion increased less severely the DNA damage sensitivity of *tel1Δ* and *sae2Δ* cells compared to *dpb4-A62S*, *dpb4Δ* cells were more sensitive than *dpb4-A62S* cells not only to a high CPT dose, but also to phleomycin and methyl methanesulfonate (MMS) (Fig. 1c). Altogether, these data suggest that the *dpb4-A62S* allele increases the DNA damage sensitivity of *tel1Δ* and *sae2Δ* cells by altering specific Dpb4 function(s).

**Dpb4 promotes DSB resection and MRX association at DSBs.** To assess the possible role of Dpb4 in DSB repair, we directly monitored ssDNA generation at the DSB ends by deleting *DPB4* or introducing the *dpb4-A62S* allele in a haploid strain carrying

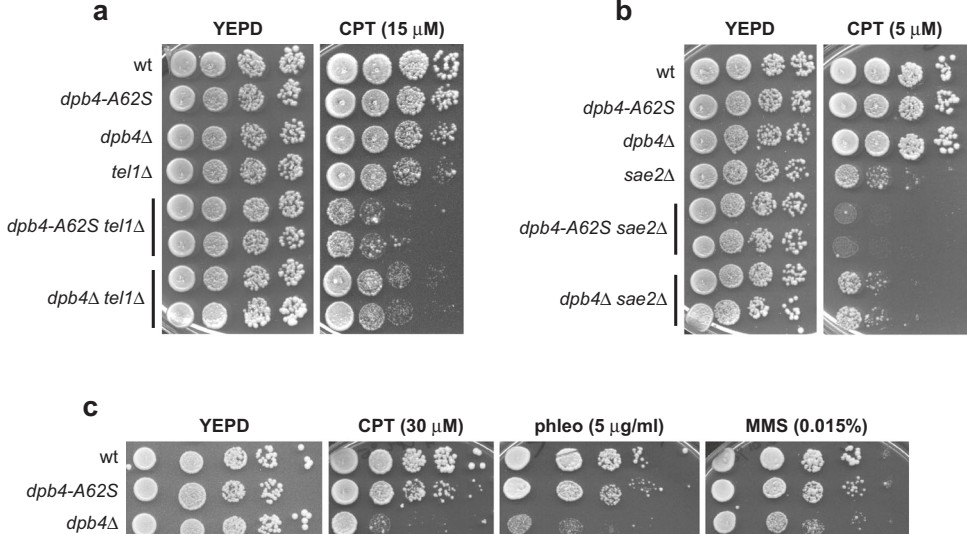

**Fig. 1 The *dpb4-A62S* mutation exacerbates the CPT sensitivity of *tel1Δ* and *sae2Δ* cells. a–c** Exponentially growing cell cultures with the indicated genotypes were serially diluted (1:10) and each dilution was spotted out onto YEPD plates with or without camptothecin (CPT), phleomycin (phleo), or methyl methanesulfonate (MMS) at the indicated concentrations.

the *HO* gene under the control of a galactose-inducible promoter. In this strain, production of the HO endonuclease by galactose addition leads to the generation at the *MAT* locus of a single DSB that cannot be repaired by HR due to the lack of the homologous donor loci *HML* and *HMR*[54]. Cells exponentially growing in raffinose were transferred to galactose to induce HO expression and genomic DNA was analyzed at different time points after HO induction. Because ssDNA is resistant to cleavage by restriction enzymes, progressively longer restriction fragments terminating at the broken end are generated as 5′–3′ nucleolytic degradation uncovers one after another *Ssp*I site (Supplementary Fig. 1). The progression of resection can be monitored by following the kinetics of appearance of these longer restriction fragments after denaturing gel electrophoresis and Southern blot analysis with a ssRNA probe that anneals to the unresected strand at one side of the DSB. The intensity of each resection band to the total amount of DSB products is used to measure the kinetics of resection. As previously observed[54,55], the resection products persisted throughout the experiment, suggesting that resection initiates asynchronously after HO-catalyzed DSB formation. The appearance of the ssDNA intermediates at the HO-induced DSB was less efficient in both *dpb4Δ* and *dpb4-A62S* cells compared to wild-type cells, with *dpb4Δ* cells showing the strongest resection defect (Fig. 2a, b).

The MRX complex binds rapidly to DSBs, where it initiates DSB resection[23]. Thus, we used chromatin immunoprecipitation (ChIP) and quantitative real-time PCR (qPCR) to monitor Mre11 recruitment near the HO-induced DSB in wild-type, *dpb4Δ* and *dpb4-A62S* cells expressing fully functional Myc-tagged Mre11 (Supplementary Fig. 2a). As resection of the DSB ends has the potential to cause a 50% decrease of input DNA, the ChIP signals were normalized not only to the efficiency of DSB induction, but also to the corresponding input for each time point. Mre11 association to the HO-induced DSB was lower in *dpb4Δ* and *dpb4-A62S* cells than in wild-type cells (Fig. 2c), with *dpb4Δ* cells again showing the strongest reduction. This decreased Mre11 recruitment was not due to lower Mre11 protein levels, as similar Mre11 amounts could be detected in protein extracts from wild-type, *dpb4Δ*, and *dpb4-A62S* cells (Fig. 2d). The reduction of Mre11 association at DSBs correlates with the severity of the DSB resection defect displayed by *dpb4Δ* and *dpb4-A62S* cells,

suggesting that the decreased Mre11 persistence at DSBs can account for the resection defect displayed by these mutants.

**Dpb4 promotes histone removal near DSBs.** DSB repair occurs within a chromatin context, in which DNA is packaged into nucleosomes. The density of nucleosome packaging has the potential to influence DSB repair and is regulated by ATP-dependent chromatin remodelers, which use the energy derived from ATP hydrolysis to evict, assemble, reposition or exchange histones throughout the genome[46]. Chromatin immunoprecipitation experiments indicate that nucleosomes are removed around a DSB in both yeast and mammalian cells, supporting the hypothesis that nucleosomes represent barriers to nuclease activity[15–21,56,57].

Dpb4 is part of the chromatin-remodeling ISW2/hCHRAC complex[40,44,45], which catalyzes nucleosome sliding[58–60]. Thus, we asked whether the poor Mre11 association and the resection defect displayed by *dpb4Δ* and *dpb4-A62S* cells are due to nucleosome retention at the DSB ends. We used ChIP analysis and qPCR to evaluate histone H2A and H3 occupancy centromere-proximal to the irreparable HO-induced DSB at the *MAT* locus. To exclude possible effects of DNA replication on histone association to DNA, HO expression was induced by galactose addition to G2-arrested cells that were kept arrested in G2 with nocodazole. As expected, H2A and H3 signals near the HO-induced DSB decreased in wild-type cells, while they remained high in both *dpb4Δ* and *dpb4-A62S* cells, with *dpb4Δ* cells showing the strongest removal defect (Fig. 2e). Taken together, these findings indicate that Dpb4 is required for nucleosome removal at DSBs, suggesting that the lack of this function can account for both the poor MRX association at DSBs and the delay of DSB resection of *dpb4Δ* and *dpb4-A62S* cells.

**Dpb4 promotes checkpoint activation in response to DSBs.** The lack of Dpb4 impairs DSB resection, Mre11 association at DSBs, and histone removal more severely than the presence of the Dpb4-A62S mutant variant (Fig. 2). However, Dpb4-A62S exacerbates the DNA damage sensitivity of both *tel1Δ* and *sae2Δ* cells more severely than *DPB4* deletion (Fig. 1a, b), suggesting that the synthetic effects caused by Dpb4-A62S are due to

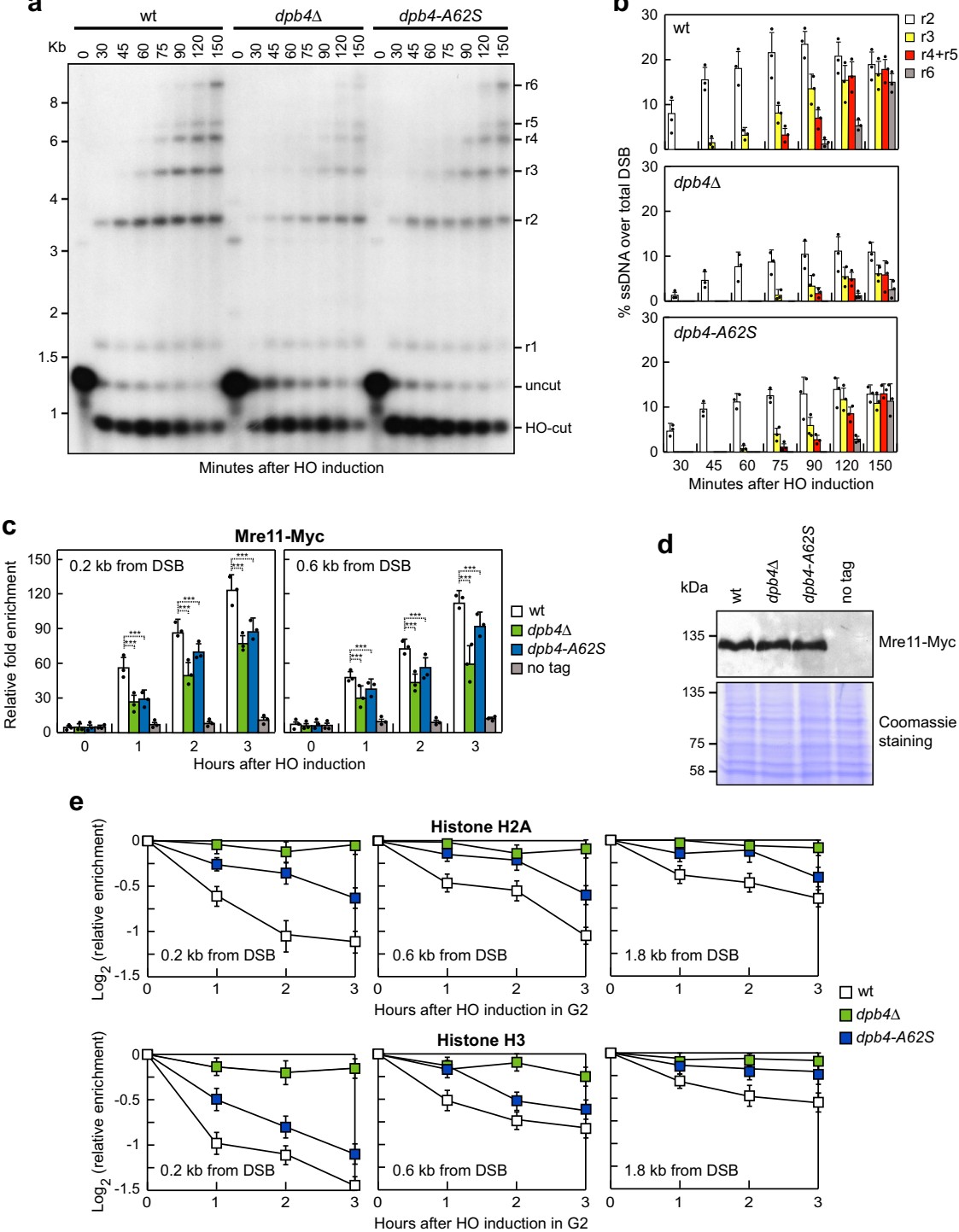

**Fig. 2 dpb4Δ and dpb4-A62S alleles reduce DSB resection, MRX association at DSBs, and histone removal from the DSB ends. a** DSB resection. YEPR exponentially growing cell cultures of JKM139 derivative strains, carrying the HO cut site at the *MAT* locus, were transferred to YEPRG at time zero. *Ssp*I-digested genomic DNA was hybridized with a single-stranded *MAT* probe that anneals with the unresected strand. 5′–3′ resection produces *Ssp*I fragments (r1 through r6) detected by the probe. **b** Densitometric analysis of the resection products. The mean values of three independent experiments as in **a** are represented with error bars denoting standard deviation (s.d.). **c** Exponentially growing YEPR cell cultures of JKM139 derivative strains were transferred to YEPRG to induce HO. Relative fold enrichment of Mre11-Myc at the HO-induced DSB was evaluated after ChIP with anti-Myc antibody and qPCR. The mean values of three independent experiments are represented with error bars denoting s.d. ***p < 0.005 (unpaired two-tailed Student's *t*-test). **d** Western blot with anti-Myc antibodies of protein extracts from exponentially growing cells. This experiment was performed independently three times with similar results. **e** HO expression was induced at time zero by galactose addition to G2-arrested cells that were kept arrested in G2 by nocodazole throughout the experiment. Relative fold enrichment of H2A or H3 at the HO-induced DSB was evaluated after ChIP with anti-H2A or anti-H3 antibody and qPCR analysis. The mean values of three independent experiments are represented with error bars denoting s.d. See source data file for statistical analysis.

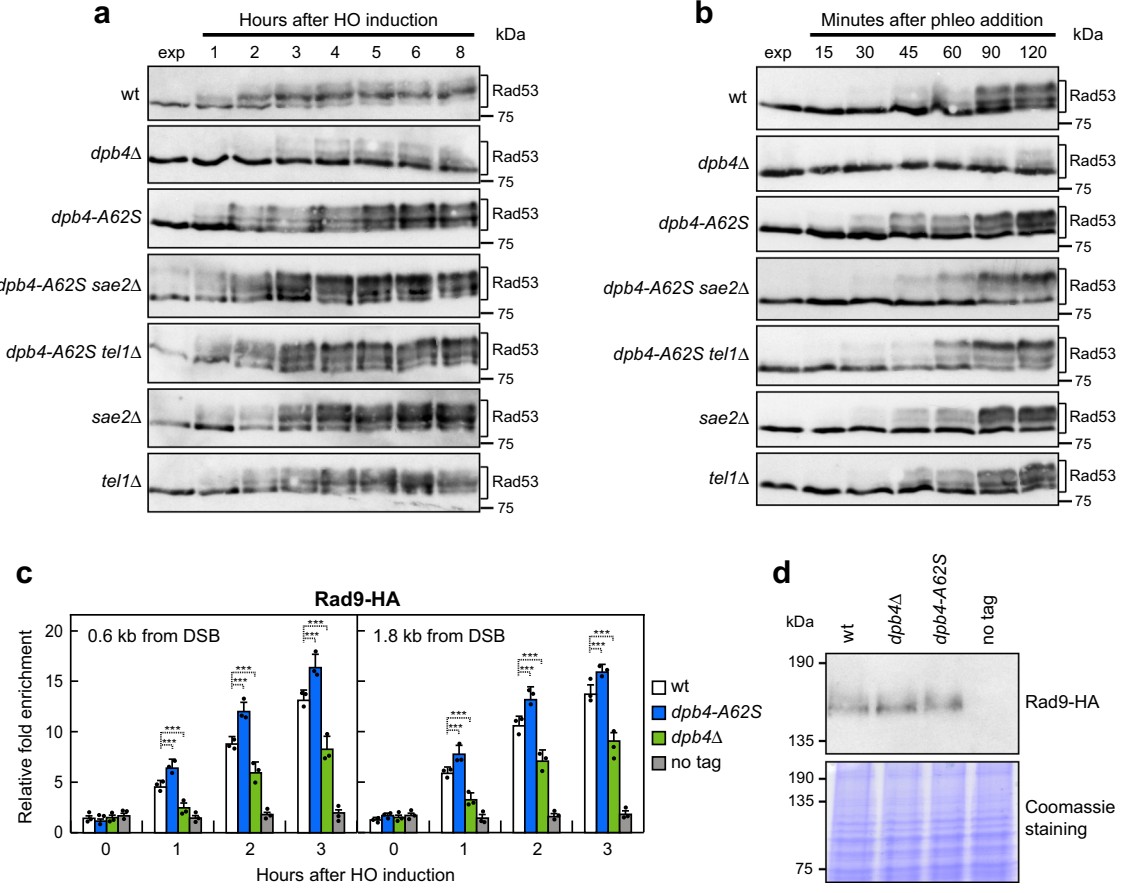

**Fig. 3 Opposite effect of _dpb4Δ_ and _dpb4-A62S_ on checkpoint activation and Rad9 association at DSBs. a** YEPR exponentially growing cell cultures of JKM139 derivative strains were transferred to YEPRG at time zero. Western blot analysis with anti-Rad53 antibodies of protein extracts from samples taken at the indicated times after HO induction. The experiment was performed independently two times with similar results. **b** Phleomycin (10 μg/mL) was added to exponentially growing cells followed by western blot analysis with anti-Rad53 antibodies. The experiment was performed independently four times with similar results. **c** Exponentially growing YEPR cell cultures of JKM139 derivative strains were transferred to YEPRG to induce HO expression. Relative fold enrichment of Rad9-HA at the HO-induced DSB was evaluated after ChIP with anti-HA antibody and qPCR. The mean values of three independent experiments are represented with error bars denoting s.d. ***$p < 0.005$ (unpaired two-tailed Student's _t_-test). **d** Western blot with anti-HA antibodies of protein extracts from exponentially growing cells. The experiment was performed independently three times with similar results.

changes of Dpb4 function in cellular processes other than DSB resection.

DSB formation leads to the activation of a checkpoint response that depends primarily on Mec1, which promotes activation of the Rad53 effector kinase[61]. Rad9 links the signal transduction from Mec1 to Rad53 by acting as a scaffold to allow Rad53 intermolecular autophosphorylation and activation[24–26]. We measured checkpoint activation in _dpb4Δ_ and _dpb4-A62S_ cells after HO-induced DSB formation or phleomycin treatment, by following Rad53 phosphorylation that is required for activation of Rad53 as a kinase and is detectable as a decrease of its electrophoretic mobility. When HO was induced by galactose addition to exponentially growing cells, the amount of slowly migrating phosphorylated Rad53 was much lower in _dpb4Δ_ cells than in wild type (Fig. 3a). Similar results were obtained also when exponentially growing cells were treated with the radio-mimetic drug phleomycin (Fig. 3b), indicating that Dpb4 is required to activate a checkpoint in response to DSBs. By contrast, the amount of the slowest migrating Rad53 form was slightly higher in _dpb4-A62S_ cells than in wild-type cells and further increased in _dpb4-A62S tel1Δ_ and _dpb4-A62S sae2Δ_ cells compared to _dpb4-A62S_, _tel1Δ_, and _sae2Δ_ cells both after HO

induction (Fig. 3a) and phleomycin addition (Fig. 3b). Thus, the lack of Dpb4 reduces Rad53 activation in response to DSBs, whereas the Dpb4-A62S mutant variant enhances it.

As Dpb4 was shown to facilitate the association of the checkpoint protein Rad9 with telomeres by an unknown mechanism[62], we analyzed Rad9 association at the HO-induced DSB by ChIP analysis and qPCR in cells expressing fully functional HA-tagged Rad9 (Supplementary Fig. 2b). Rad9 association to the HO-induced DSB was decreased in _dpb4Δ_ cells compared to wild type, while it was increased in _dpb4-A62S_ cells (Fig. 3c), although similar Rad9 amounts could be detected in protein extracts prepared from wild-type, _dpb4Δ_ and _dpb4-A62S_ cells (Fig. 3d). These findings indicate that Dpb4 promotes Rad9 association at DSBs and checkpoint activation, and that this Dpb4 checkpoint function is enhanced by the _dpb4-A62S_ mutation that leads to an increased Rad9 persistence at DSBs.

The finding that the _dpb4-A62S_ allele increases Rad53 activation raises the possibility that the severe DNA damage hypersensitivity of _dpb4-A62S tel1Δ_ and _dpb4-A62S sae2Δ_ cells compared to _tel1Δ_ and _sae2Δ_ cells might be due to the hyperactivation of the checkpoint response. If this were the case, either _RAD9_ deletion or expression of the kinase defective

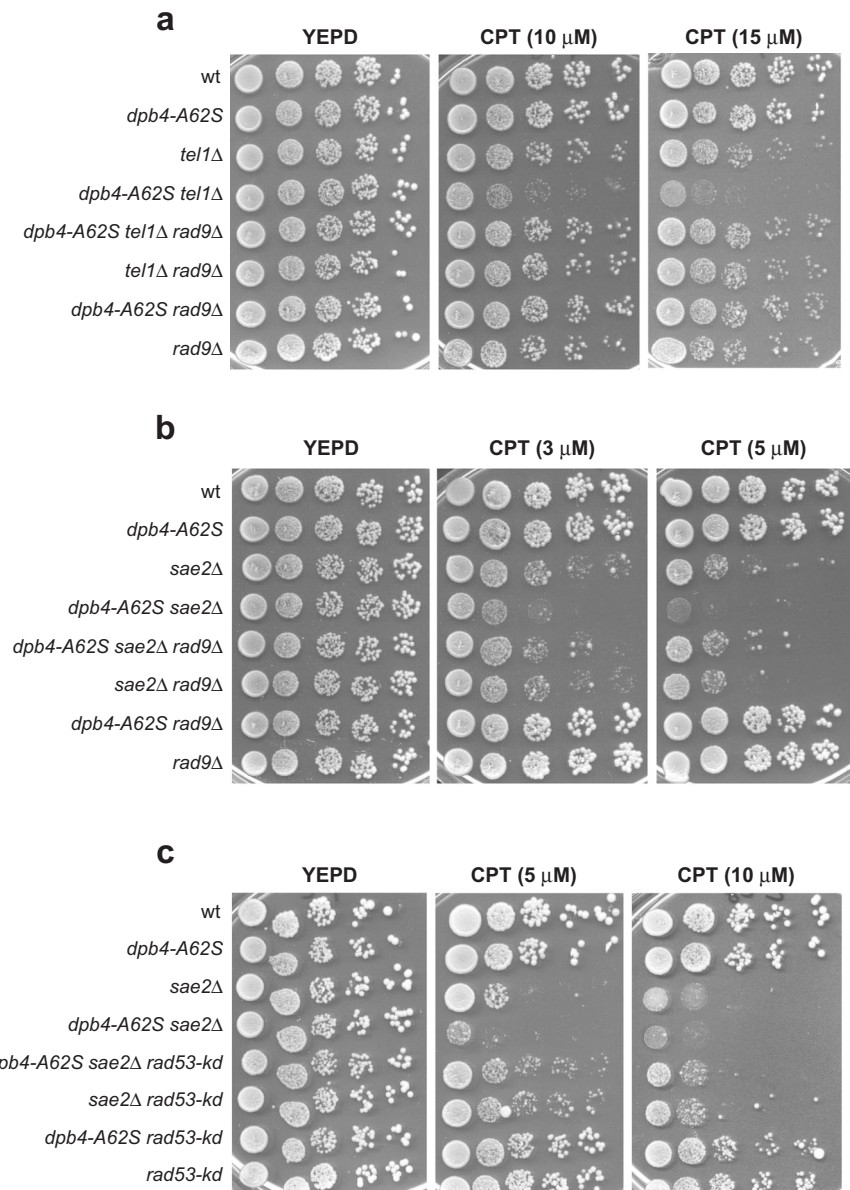

**Fig. 4 Dpb4-A62S increases the DNA damage sensitivity of *tel1Δ* and *sae2Δ* cells in a Rad9- and Rad53-dependent manner. a–c** Exponentially growing cultures with the indicated genotypes were serially diluted (1:10) and each dilution was spotted out onto YEPD plates with or without CPT.

*rad53-K227A* (*rad53-kd*) allele should decrease the DNA damage hypersensitivity of *dpb4-A62S tel1Δ* and *dpb4-A62S sae2Δ* cells. Indeed, *RAD9* deletion suppressed the CPT hypersensitivity of *dpb4-A62S tel1Δ* cells, as *dpb4-A62S tel1Δ rad9Δ* cells were as sensitive to CPT as *tel1Δ rad9Δ* cells (Fig. 4a). Furthermore, *RAD9* deletion was epistatic to *dpb4-A62S* with respect to the CPT sensitivity of *sae2Δ* cells. In fact, *dpb4-A62S sae2Δ rad9Δ* cells, which were less sensitive to DNA damaging agents than *dpb4-A62S sae2Δ* cells, were as sensitive as *sae2Δ rad9Δ* cells (Fig. 4b).

Unfortunately, due to the poor viability of *rad53-kd tel1Δ* double mutant even in the absence of DNA damaging agents, we could not evaluate the effect of the *rad53-kd* allele on *dpb4-A62S tel1Δ* cells. In any case, expression of *rad53-kd*, which partially suppressed the DNA damage sensitivity of *sae2Δ* cells[63], was epistatic to *dpb4-A62S* with respect to the CPT sensitivity of *dpb4-A62S sae2Δ* cells, as *dpb4-A62S sae2Δ rad53-kd* cells were as sensitive to CPT as *sae2Δ rad53-kd* cells (Fig. 4c). Thus, Rad9 and Rad53 kinase activity are required for Dpb4-A62S to increase the

DNA damage sensitivity of *tel1Δ* and *sae2Δ* cells, suggesting that the enhanced checkpoint activation by Dpb4-A62S leads to DNA damage-induced lethality in the presence of unrepaired DNA lesions.

**Dpb4 and Dot1 promote Rad9 association to DSBs by acting in the same pathway and independently of Dpb11 and γH2A.** In both yeast and mammals, Rad9 association with chromatin involves at least three pathways. First, Rad9 is constitutively bound to chromatin even in the absence of DNA damage through an interaction with histone H3 methylated at Lys79 (H3-K79me), a modification that is catalyzed by the methyltransferase Dot1[64–69]. Furthermore, phosphorylation of Ser462 and Thr474 Rad9 residues by Cdk1 leads to Rad9 interaction with Dpb11[32,70], which is recruited to DSBs by the 9-1-1 complex[29,71]. Finally, DNA damage induces Rad9 binding to histone H2A that has been phosphorylated at Ser129 (γH2A) by the checkpoint kinases Mec1/ATR and Tel1/ATM[55,72–74].

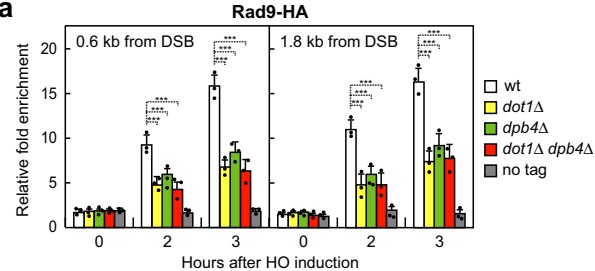

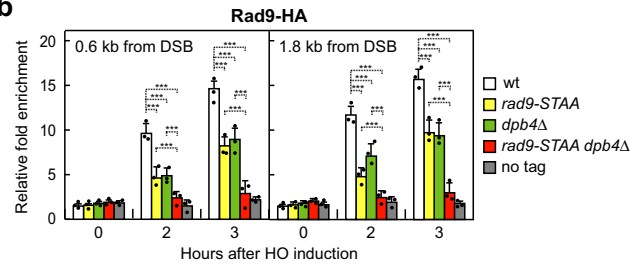

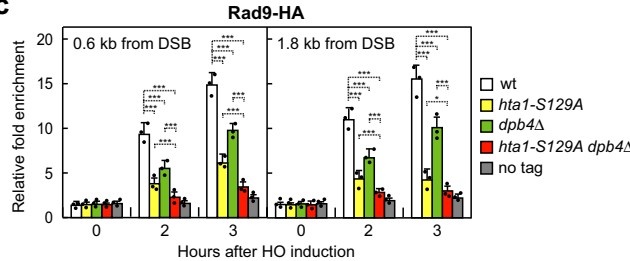

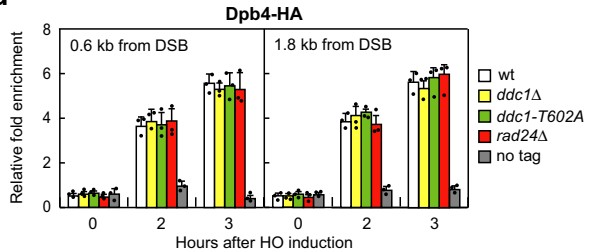

**Fig. 5 Dpb4 promotes Rad9 association at DSBs independently of γH2A and Dpb11–Rad9 interaction and by acting in the same pathway of Dot1. a–d** Exponentially growing YEPR cell cultures of JKM139 derivative strains were transferred to YEPRG to induce HO expression. Relative fold enrichment of Rad9-HA (**a–c**) and Dpb4-HA (**d**) at the HO-induced DSB at the *MAT* locus was evaluated after ChIP with anti-HA antibody and qPCR. Strains carrying the *hta1-S129A* allele also contain the deletion of the *HTA2* gene. The mean values of three independent experiments are represented with error bars denoting s.d. ***$p < 0.005$; *$p < 0.05$ (unpaired two-tailed Student's *t*-test).

To investigate whether Dpb4 promotes Rad9 association to DSBs by acting in one of the above pathways, we analyzed the contribution of Dpb4 in supporting Rad9 association to DSBs in cells that were defective in Rad9 binding to either H3-K79me, γH2A, or Dpb11. As expected, the lack of Dot1, which abolishes H3-K79me generation[64–66], decreased Rad9 association to the HO-induced DSB (Fig. 5a). A similar decrease of Rad9 persistence at DSBs could be detected upon expression of either the *rad9-S462A, T474A* (*rad9-STAA*) (Fig. 5b), or the *hta1-S129A* allele (Fig. 5c), which abolish Rad9–Dpb11 interaction and γH2A generation, respectively. Interestingly, *DPB4* deletion did not further decrease the amount of Rad9 bound to DSBs in *dot1Δ* cells (Fig. 5a), indicating that Dpb4 and Dot1 promote Rad9

association at DSBs by controlling the same pathway. By contrast, Rad9 association at DSBs was markedly decreased in both *rad9-STAA dpb4Δ* (Fig. 5b) and *hta1-S129A dpb4Δ* (Fig. 5c) double mutants compared to each single mutant, indicating that Dpb4 function in promoting Rad9 association at DSBs occurs independently of Rad9–γH2A and Dpb11–Rad9 interactions. Consistent with this conclusion, when HO was induced in exponentially growing cells expressing fully functional Dpb4-HA-tagged protein (Supplementary Fig. 2c), Dpb4 recruitment to the HO-induced DSB requires neither the 9-1-1 complex nor the interaction between 9-1-1 and Dpb11. In fact, the lack of Ddc1 or the presence of the *ddc1-T602A* allele, which specifically abrogates 9-1-1 binding to Dpb11[31], did not decrease Dpb4 association to the HO-induced DSB (Fig. 5d).

### Different interactors support Dpb4 functions in DSB resection and checkpoint activation

The Dpb4 protein is shared by two highly conserved protein complexes: the chromatin-remodeling ISW2[40,44,45] and the Pol ε complexes[33–35]. In both complexes, Dpb4 forms a dimer that resembles H2A–H2B by interacting with two different histone fold proteins: Dls1 in the ISW2 complex and Dpb3 in the Pol ε complex[35,40,45].

Both Pol2 and Dpb2 subunits of the Pol ε complex are essential for cell viability. We, therefore, investigated the effects of deleting *DPB3*, *DLS1*, and the ATPase encoding gene *ISW2* in order to assess the contribution of Pol ε and ISW2 complexes in supporting Dpb4 functions in DSB resection and checkpoint activation. Deletion of *ISW2* and *DLS1*, but not of *DPB3*, severely reduced removal of H2A (Fig. 6a) and H3 histones (Supplementary Fig. 3) from the HO-induced DSB. Furthermore, *isw2Δ* and *dls1Δ* cells showed decreased Mre11 association to the HO-induced DSB, whereas *dpb3Δ* cells did not (Fig. 6b). Consistent with the finding that Isw2 is the catalytic subunit, whereas both Dpb4 and Dls1 help nucleosome sliding by ISW2[59], *isw2Δ* cells showed a more severe impairment of Mre11 association to the HO-induced DSB compared to both *dpb4Δ* and *dls1Δ* cells (Fig. 6b). *DPB4* deletion did not further decrease the amount of Mre11 bound at DSB in *isw2Δ* cells (Fig. 6b), indicating that Dpb4 and Isw2 promote MRX association by acting in the same pathway. Finally, both *isw2Δ* and *dls1Δ* cells were defective in resection of the HO-induced DSB compared to wild-type cells (Fig. 6c, d). Altogether, these findings indicate that Dpb4 promotes histone removal, MRX association to DSBs, and DSB resection by acting in the ISW2 complex.

It has been proposed that Dpb4 acts as an anchor point on DNA for Isw2[75], prompting us to test whether the defect in histone removal in *dpb4Δ* cells is due to a decreased association of the Isw2 catalytic subunit to DSBs. The amount of Isw2 bound at the HO-induced DSB was markedly reduced in *dpb4Δ* cells (Fig. 6e), although similar Isw2 levels were present in both wild-type and *dpb4Δ* cell extracts (Fig. 6f), indicating that Dpb4 promotes Isw2 association to DSBs. By contrast, *dpb4-A62S* cells, which showed defective nucleosome eviction from DSBs (Fig. 2e), exhibited increased Isw2 persistence at the HO-induced DSB (Fig. 6e), suggesting that not only a decreased but also an increased Isw2 association at DSBs might impair histone removal. Consistent with this hypothesis, deletion of the negatively charged C terminus of the *D. melanogaster* Dpb4 ortholog enhances DNA binding but inhibits nucleosome sliding[59].

In agreement with the conclusion that Dpb3 did not support Dpb4 function in removing histones and in promoting Mre11 association to DSBs (Fig. 6a, b), *dpb3Δ* cells were not defective in DSB resection (Fig. 7a and Supplementary Fig. 4). Instead, Dpb3, but not the ISW2 complex, supports the Dpb4 function in checkpoint activation. In fact, *dpb3Δ* cells, but not *isw2Δ* cells,

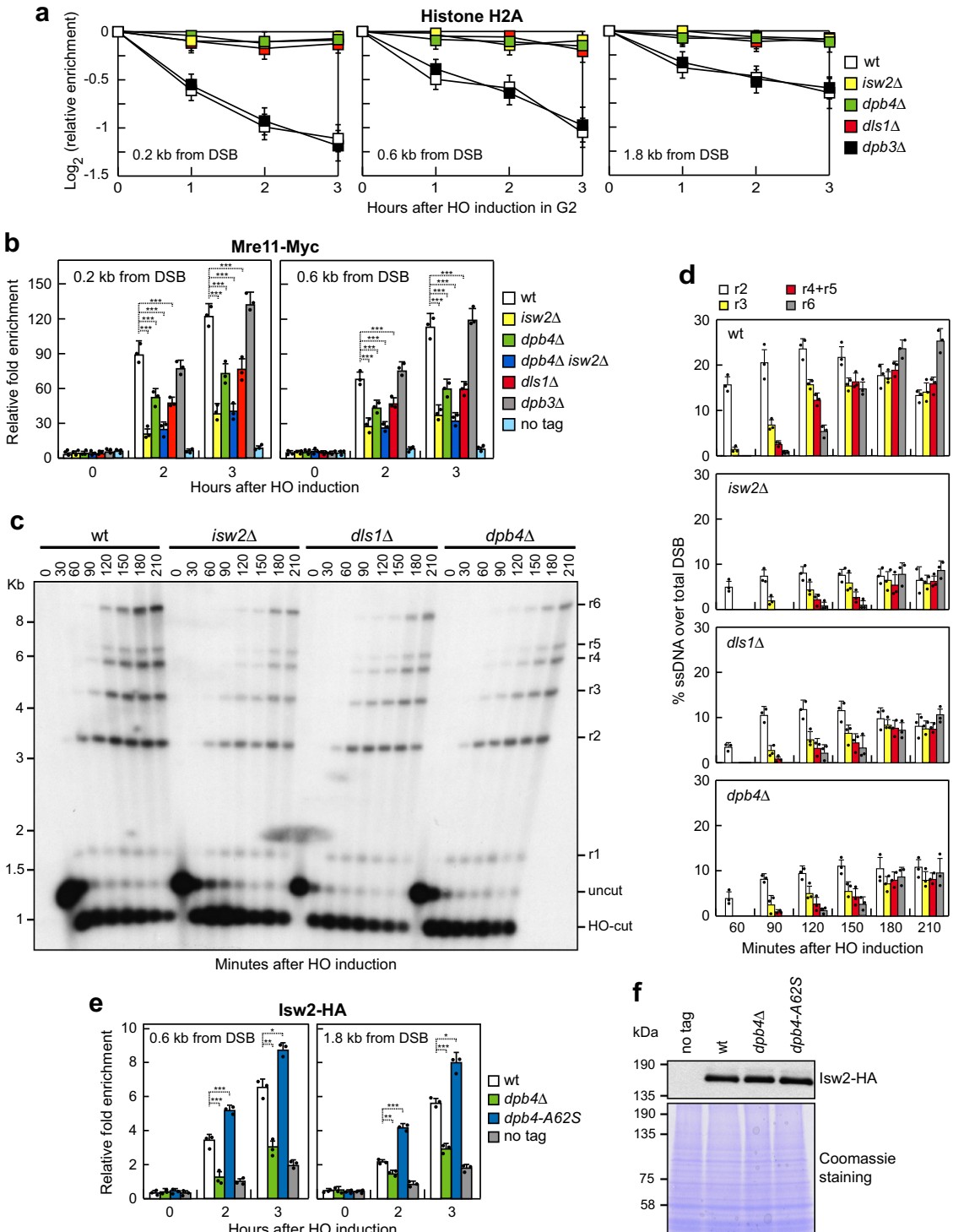

showed decreased Rad53 phosphorylation after HO induction (Fig. 7b) or phleomycin treatment (Fig. 7c). Similarly, *dpb3Δ* cells, but not *isw2Δ* cells, showed reduced Rad9 association at the HO-induced DSB compared to wild-type cells (Fig. 7d). The lack of Dpb4 did not further decrease the amount of Rad9 bound at DSB in *dpb3Δ* cells, indicating that Dpb3 and Dpb4 act in the same pathway to promote Rad9 association to DSBs (Fig. 7d). Consistent with the finding that the ISW2 complex is not involved in checkpoint activation, Isw2 and Dls1 proteins are not required to increase the DNA damage sensitivity of *dpb4-A62S tel1Δ* cells, as *dpb4-A62S dls1Δ tel1Δ* and *dpb4-A62S isw2Δ tel1Δ*

cells were as sensitive to CPT as *dpb4-A62S tel1Δ* cells (Fig. 7e, f). Altogether, these findings indicate that Dpb4 acts in the ISW2 complex to promote MRX association at DSBs and DSB resection, whereas it acts with Dpb3 to promote checkpoint activation.

The Dpb3–Dpb4 heterodimer is part of the Pol ε holoenzyme[35], which was previously shown to promote checkpoint activation in response to DNA replication stress[76–79]. The Dpb3–Dpb4 complex was shown to enhance both the processivity of Pol ε[42,43,80] and the DNA binding activity of Pol2[41]. The Pol ε checkpoint function relies on the C-terminal domain of Pol2, which is essential for cell viability, making it difficult to assess

**Fig. 6 The lack of Dls1 and Isw2, but not of Dpb3, impairs histone removal, MRX association to DSBs, and DSB resection. a** HO expression was induced at time zero by galactose addition to G2-arrested cells that were kept arrested in G2 by nocodazole. Relative fold enrichment of H2A at the HO-induced DSB was evaluated after ChIP with an anti-H2A antibody. The mean values of three independent experiments are represented with error bars denoting s.d. See source data file for statistical analysis. **b** Exponentially growing YEPR cell cultures were transferred to YEPRG to induce HO expression. Relative fold enrichment of Mre11-Myc at the HO-induced DSB was evaluated after ChIP with anti-Myc antibody and qPCR. The mean values of three independent experiments are represented with error bars denoting s.d. ***$p < 0.005$ (unpaired two-tailed Student's $t$-test). **c** DSB resection. YEPR exponentially growing cell cultures were transferred to YEPRG at time zero to induce HO production. $Ssp$I-digested genomic DNA was analyzed as in Fig. 2a. **d** Densitometric analysis of the resection products. The mean values of three independent experiments as in (**c**) are represented with error bars denoting s.d. See source data file for statistical analysis that was performed using unpaired two-tailed Student's $t$-test. **e** Exponentially growing YEPR cell cultures of JKM139 derivative strains were transferred to YEPRG to induce HO expression. Relative fold enrichment of Isw2-HA at the HO-induced DSB was evaluated after ChIP with anti-HA antibody and qPCR. The mean values of three independent experiments are represented with error bars denoting s.d. ***$p < 0.005$; **$p < 0.01$; *$p < 0.05$ (unpaired two-tailed Student's $t$-test). **f** Western blot with anti-HA antibodies of protein extracts from exponentially growing cells. The experiment was performed independently three times with similar results.

whether Dpb3–Dpb4 promotes Rad9 association to DSBs and checkpoint activation by acting within the Pol ε complex. As the enhanced checkpoint activation caused by Dpb4$^{A62S}$ is likely due to the increased Rad9 association to DSBs, if the Dpb4 checkpoint function involves the Pol ε holoenzyme, Dpb4$^{A62S}$ should cause an increased Pol2 persistence to DSBs. To exclude possible effects of DNA replication, HO expression was induced by galactose addition to G2-arrested cells that were kept arrested in G2 with nocodazole. Following HO induction, Pol2 was recruited to the HO-induced DSB (Fig. 7g). Furthermore, although wild-type, *dpb4Δ*, and *dpb4-A62S* cells contained a similar amount of Pol2 (Fig. 7h), the A62S mutation increased Pol2 occupancy at the HO-induced DSB, whereas the lack of Dpb4 decreased it (Fig. 7g), suggesting that Dpb4 might act through Pol ε to promote checkpoint activation in response to DSBs.

**The A62S mutation favors the formation of high order Dpb4–Dpb3 and Dpb4–Dls1 complexes on DNA**. Dpb4, Dpb3, and Dls1 contain a histone fold (helix–turn–helix–turn–helix) domain (Fig. 8a), through which they interact to form H2A–H2B-like Dpb3–Dpb4 and Dls1–Dpb4 heterodimers[40,44,45,81]. Sequence and structural analyses indicate that the A62 residue is localized on the α2 helix within the histone fold domain and interacts with I74 and I87 residues that are localized on the α3 helix of the histone fold domain of Dpb3 and Dls1, respectively (Fig. 8b).

The A62S mutation did not impair Dpb3–Dpb4 and Dls1–Dpb4 complex formation in vivo. In fact, when Dpb4-HA or Dpb4$^{A62S}$-HA was immunoprecipitated with anti-HA antibodies, a similar amount of Dpb3-Myc could be detected in the immunoprecipitates (Supplementary Fig. 5a). Similarly, when Dls1-HA was immunoprecipitated with anti-HA antibodies, similar amounts of Dpb4-Myc and Dpb4$^{A62S}$-Myc could be detected in the immunoprecipitates (Supplementary Fig. 5b).

The lack of Dpb4 impairs the association to DSBs of both Isw2 and Pol2, whereas the A62S mutation increases it, prompting us to test whether the functions of Dpb4 in both histone eviction and checkpoint activation rely on its DNA binding activity that can be enhanced by the A62S mutation. When HO was induced by galactose addition to G2-arrested cells that were kept arrested in G2 with nocodazole to exclude the possible effect of DNA replication, Dpb4 was efficiently recruited close to the HO cut site (Fig. 8c). Although the A62S substitution did not affect Dpb4 protein level within cells (Fig. 8d) and the addition of the HA tag at the Dpb4$^{A62S}$ C terminus did not alter the DNA damage sensitivity of *dpb4-A62S* cells (Supplementary Fig. 2c), the amount of Dpb4$^{A62S}$ bound at the HO-induced DSB was higher than that of wild-type Dpb4 (Fig. 8c). This finding suggests that the A62S mutation increases the Dpb4 association to dsDNA.

To investigate whether the A62S mutation enhances Dpb4 occupancy at DSBs by increasing the DNA binding affinity of Dpb3–Dpb4 and/or Dls1–Dpb4 complexes, we expressed and purified Dpb3–Dpb4, Dpb3–Dpb4$^{A62S}$, Dls1–Dpb4, and Dls1–Dpb4$^{A62S}$ heterodimers as soluble protein complexes from *Escherichia coli* cells (Supplementary Fig. 6a, b). Circular dichroism (CD) spectra of Dpb3–Dpb4 and Dls1–Dpb4 present two minima at 208 nm and 222 nm, which are typical of α-helix structures (Supplementary Fig. 6c, d). The CD spectra of Dpb3–Dpb4$^{A62S}$ and Dls1–Dpb4$^{A62S}$ are similar to those of the wild-type complexes, indicating that the A62S substitution does not affect the protein secondary structure (Supplementary Fig. 6c, d). The thermal stability of the chimeric heterodimers was investigated by CD spectroscopy at a fixed wavelength (208 nm) in the 25–90 °C temperature range. The unfolding transition midpoint temperatures ($T_m$) of Dpb3–Dpb4 and Dls1–Dpb4 are 61.57 ± 0.39 and 57.05 ± 0.31 °C, respectively, whereas the A62S mutation decreases the $T_m$ of Dpb3–Dpb4 and Dls1–Dpb4 complexes of ~3 and 2 °C, respectively (Supplementary Fig. 6e, f), suggesting that the A62S mutation induces slight conformational changes in both complexes.

To test the DNA binding properties of these protein complexes, increasing concentrations of purified Dpb3–Dpb4, Dpb3–Dpb4$^{A62S}$, Dls1–Dpb4, and Dls1–Dpb4$^{A62S}$ protein complexes were incubated with a fixed amount of a 61-mer dsDNA substrate to test their ability to bind DNA in a gel electrophoretic mobility shift assay (EMSA). As previously reported[41], the addition of wild-type Dpb3–Dpb4 or Dls1–Dpb4 complexes was capable of shifting the dsDNA oligomer into a distinct slower migrating band (Fig. 8e, f), indicating that both complexes can bind dsDNA. Notably, both Dpb3–Dpb4$^{A62S}$ and Dls1–Dpb4$^{A62S}$ were capable of generating a similar slower migrating band although less efficiently compared to the corresponding wild-type complexes (Fig. 8e, f). However, they both showed the appearance of a second slower migrating band (Fig. 8e, f), suggesting that the A62S amino acid substitution favors a transition to higher-order Dpb3–Dpb4–DNA and Dls1–Dpb4–DNA complexes that can explain the increased amount of Dpb4 bound to DSBs detected by ChIP.

## Discussion

Our data show that the conserved histone fold protein Dpb4 is involved in at least two aspects of the cellular response to DSBs: (i) it promotes MRX association to DSBs, thus allowing DSB resection; (ii) it promotes Rad9 association to DSBs, thus allowing checkpoint activation. We found that Dpb4 is required to remove histones around a DSB. As the presence of nucleosomes inhibits resection both in vitro and in vivo[57,82], these data support the view that failure to remove histones from the DSB ends can

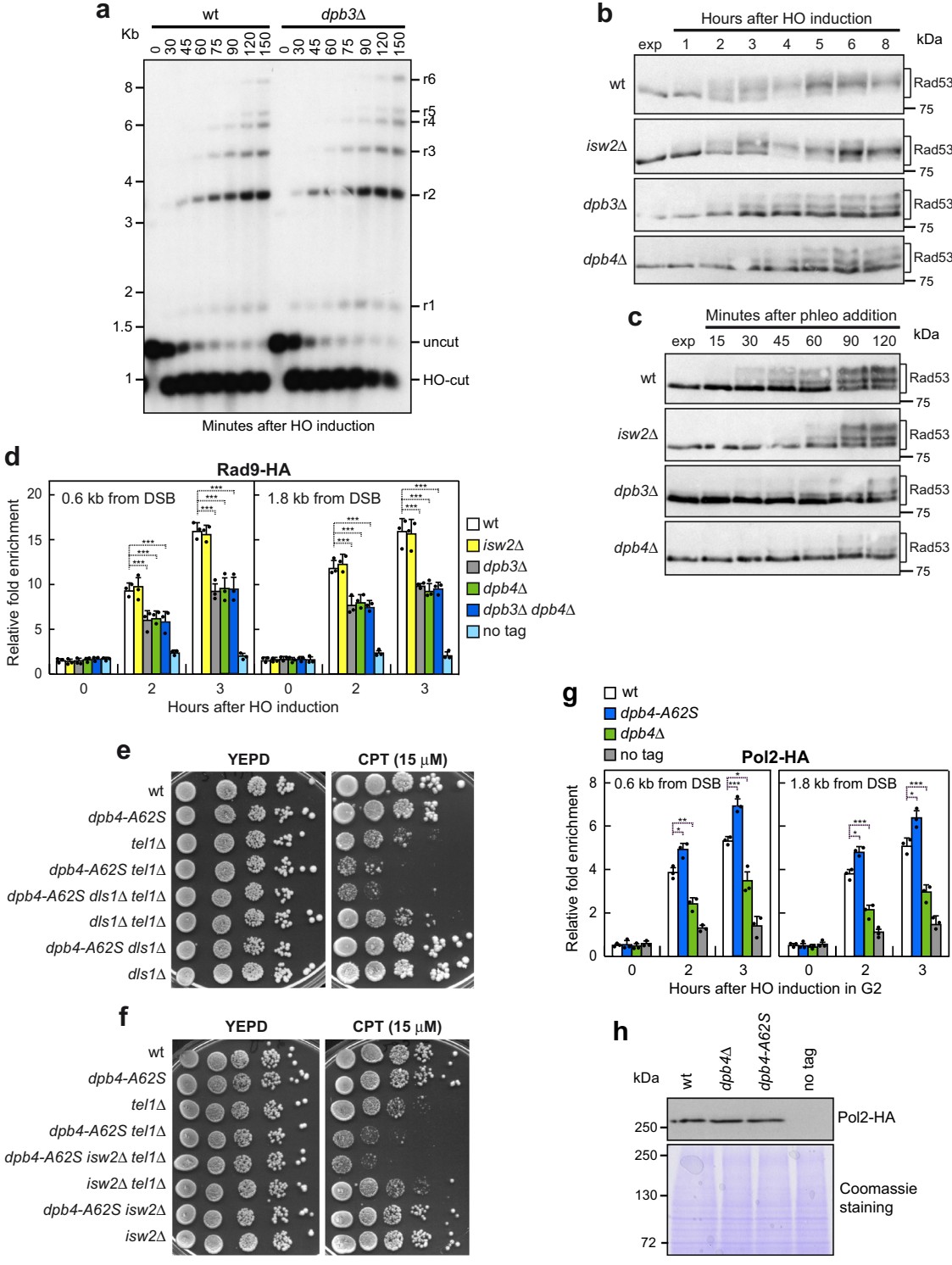

account for the decreased MRX association to DSBs and the resection defect of *dpb4Δ* cells.

Dpb4 is shared by two highly conserved protein complexes: the chromatin-remodeling ISW2/hCHRAC, which is known to slide nucleosomes by disrupting histone–DNA interactions[46,83,84], and the Pol ε, which is implicated in DNA replication and heterochromatin maintenance[40,41]. In both complexes, Dpb4 interacts with a histone fold protein, namely Dls1 in the ISW2 complex and Dpb3 in the Pol ε complex, to form H2A–H2B-like complexes[40,44,45,81].

We found that Dpb4 promotes DSB resection by acting with Dls1 and Isw2 subunits of the ISW2 complex, whereas it facilitates Rad9 association to DSBs and checkpoint activation by acting with Dpb3. In fact, similar to *DPB4* deletion, the lack of Dls1 or Isw2, but not of Dpb3, severely reduces histone removal from the DSB ends, MRX association to DSBs and DSB resection. By contrast, the lack of Isw2 does not reduce checkpoint activation and Rad9 association to DSBs. Rather, the Dpb4 checkpoint function relies on Dpb4 interaction with Dpb3, whose lack reduces Rad9 association to DSBs and DSB-induced Rad53

**Fig. 7 The lack of Dpb3, but not of Isw2, impairs Rad9 association to DSB and checkpoint activation. a** DSB resection. YEPR exponentially growing cell cultures were transferred to YEPRG at time zero to induce HO production. SspI-digested genomic DNA was analyzed as in Fig. 2a. The experiment was performed independently three times with similar results; see Supplementary Figure 4 for the densitometric analysis of the resection products from three independent experiments as in **a**. **b** YEPR exponentially growing cell cultures of JKM139 derivative strains were transferred to YEPRG at time zero to induce HO. Protein extracts were subjected to western blot analysis with anti-Rad53 antibodies. **c** Phleomycin (10 μg/mL) was added to exponentially growing cells followed by western blot analysis with anti-Rad53 antibodies. The experiments in **b** and **c** were performed independently two times with similar results. **d** Exponentially growing YEPR cell cultures of JKM139 derivative strains were transferred to YEPRG to induce HO expression. Relative fold enrichment of Rad9-HA at the HO-induced DSB was evaluated after ChIP with anti-HA antibody and qPCR. The mean values of three independent experiments are represented with error bars denoting s.d. ***$p < 0.005$ (unpaired two-tailed Student's $t$-test). **e, f** Serial dilutions of exponentially growing cultures onto YEPD plates with or without CPT. **g** HO expression was induced at time zero by galactose addition to G2-arrested cells that were kept arrested in G2 by nocodazole. Relative fold enrichment of Pol2-HA at the HO-induced DSB was evaluated after ChIP with anti-HA antibody and qPCR. The mean values of three independent experiments are represented with error bars denoting s.d. ***$p < 0.005$; **$p < 0.01$; *$p < 0.05$ (unpaired two-tailed Student's $t$-test). **h** Western blot with anti-HA antibodies of protein extracts from G2-arrested cells. The experiment was performed independently three times with similar results.

phosphorylation. The lack of Dpb4 does not further reduce Rad9 persistence at DSBs in $dpb3\Delta$ cells, indicating that Dpb3 and Dpb4 promote Rad9 association to DSBs by acting in the same pathway. The Dpb4-mediated histone removal and Rad9 loading to DSBs occurs independently to each other, as the lack of Isw2 impairs histone removal from DSBs but not Rad9 association to DSBs, whereas the lack of Dpb3 impairs Rad9 association to DSBs but not histone removal from DSBs. Furthermore, the finding that $dpb3\Delta$ cells, but not $isw2\Delta$ cells, showed reduced Rad53 activation indicates that the checkpoint defect of $dpb4\Delta$ cells is caused by the decreased Rad9 association at DSBs, rather than by the reduced amount of ssDNA caused by defective resection.

The function of Dpb4 in promoting Isw2 and Rad9 association to DSBs is enhanced by the A62S mutation that leads to an increased Dpb4 persistence at DSBs, suggesting that the Dpb4 functions in both chromatin-remodeling and checkpoint activation rely on its DNA binding property. Interestingly, although the A62S mutation slightly reduces the DNA binding affinity of both Dpb3–Dpb4 and Dls1–Dpb4 complexes, it favors the formation of higher-order DNA–Dpb3–Dpb4 and DNA–Dls1–Dpb4 complexes. Although the nature of these complexes requires further investigation, their formation suggests that the increased amount of chromatin-bound Dpb4$^{A62S}$ detected by ChIP is due to a transition to high stoichiometry protein–DNA complexes.

The Dpb4$^{A62S}$ mutant variant enhances Isw2 association to DSBs but reduces histone removal from the DSB ends. Similarly, deletion of the negatively charged C terminus of the *D. melanogaster* Dpb4 ortholog enhances DNA binding but inhibits nucleosome sliding[59], suggesting that not only a poor, but also an increased Isw2 persistence to DSBs impairs Isw2 activity. As nucleosome mobilization by Isw2 involves DNA translocation inside the nucleosome that requires Isw2 ability to break and reform DNA-histone contacts[85], increasing the interaction between Isw2 and the nucleosomal DNA might enhance the energetic barrier to nucleosome repositioning, thus explaining the histone removal defect of *dpb4-A62S* cells.

The Dpb3–Dpb4 complex is flexibly tethered to the core subunits of Pol ε[35], which was previously shown to activate a checkpoint in response to DNA replication stress independently of the 9-1-1 complex[76–79]. This finding raises the question of whether Dpb3–Dpb4 dimer acts through Pol ε to activate the checkpoint. We found that Pol2 is recruited to DSBs independently of DNA replication and Dpb4$^{A62S}$ leads to an increased Pol2 association at DSBs, suggesting that Dpb4 might act within the Pol ε holoenzyme to enhance Rad9 association to DSBs and checkpoint activation.

One possibility is that Dpb4 promotes Rad9 association to DSBs by directly recruiting Rad9 to the DSB sites. However, we failed to detect any interaction between Dpb4 and Rad9 by

coimmunoprecipitation. Interestingly, Dpb4 promotes Rad9 association to DSBs by acting in the same pathway of Dot1, which is known to drive Rad9 association to DSBs by catalyzing H3-K79 methylation. H3-K79 is constitutively methylated by Dot1 in both mammalian and yeast cells[68,69,86]. Furthermore, at least in human cells, irradiation does not lead to increased histone H3-K79 methylation[67], raising the question of how DNA DSBs expose methylated H3-K79 to Rad9 recognition. The Dpb3–Dpb4 complex has been shown to bind histones H3 and H4 in the context of chromatin, and to possess intrinsic H3–H4 chaperone and DNA supercoiling activities[38,39]. Thus, it would be tempting to speculate that the Dpb3–Dpb4 complex, possibly as part of the Pol ε holoenzyme, induces re-deposition/exchange of histones H3 and H4 at the DSB ends, where subsequent H3 methylation by Dot1 would lead to exposure of histone H3 to Rad9 recognition (Fig. 8g).

In conclusion, we propose that the Dls1–Dpb4 dimer binds the DSB ends and facilitates the loading of the ISW2/hCHRAC complex, which in turn promotes MRX association to DSBs and DSB resection by sliding/removing nucleosomes from the DSB ends (Fig. 8g). The Dpb3–Dpb4 complex, in turn, promotes Rad9 association to the DSB and checkpoint activation by inducing exposure of histone H3 to Rad9 binding. Because Dpb4 is evolutionarily conserved, it will be interesting to investigate whether, depending on its interactors, it plays similar roles in DSB resection and checkpoint activation also in mammalian cells.

## Methods

**Yeast strains and media**. *S. cerevisiae* is the experimental model used in this study. Strain genotypes are listed in Supplementary Table 1. Strain JKM139, used to detect DSB resection and to perform ChIP analysis, was kindly provided by J. Haber (Brandeis University, Waltham, USA). The *ddc1-T602A* and the *rad9-STAA* alleles were kindly provided by J. Diffley (The Francis Crick Institute, London UK) and B. Pfander (Max Planck Institute of Biochemistry, Martinsried, Germany). Gene disruptions and tag fusions were generated by one-step PCR and standard yeast transformation procedure. Primers used for disruptions and gene tagging are listed in Supplementary Table 2. Cells were grown in YEP medium (1% yeast extract, 2% bactopeptone) supplemented with 2% glucose (YEPD), 2% raffinose (YEPR) or 2% raffinose and 3% galactose (YEPRG). All experiments were performed at 26 °C.

**Search for mutations that sensitize *tel1*Δ cells to CPT**. To search for mutations that sensitize *tel1*Δ cells to CPT, *tel1*Δ cells were mutagenized with ethyl methanesulfonate and plated on YEPD plates. Approximately 100000 survival colonies were replica plated on YEPD plates with or without CPT. Clones sensitive to CPT were transformed with a plasmid containing wild-type *TEL1* to identify those that lost the DNA damage hypersensitivity. The corresponding original clones were then crossed with wild-type cells to identify by tetrad analysis the clones in which the increased DNA damage sensitivity was due to the simultaneous presence of *tel1*Δ and a single-gene mutation. This mutation was identified by genome sequencing and genetic analyses. To confirm that the *dpb4-A62S* mutation was responsible for the increased DNA damage sensitivity of *tel1*Δ cells, a *KANMX* gene was integrated downstream of the *dpb4-A62S* stop codon and the resulting

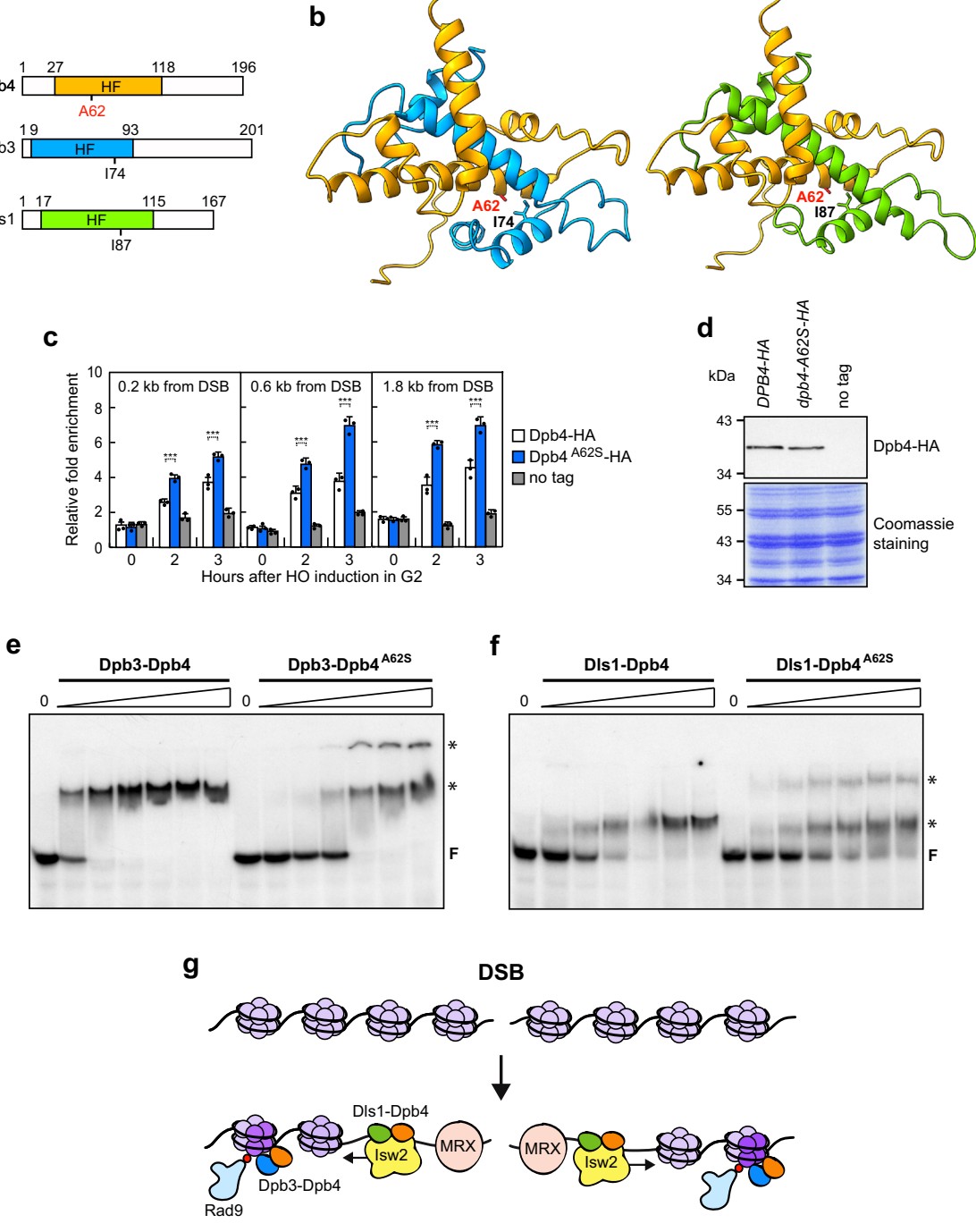

**Fig. 8 Effect of the A62S mutation on the DNA binding properties of Dpb3–Dpb4 and Dls1–Dpb4 heterodimers. a** Representation of Dpb4, Dpb3, and Dls1 proteins. The histone fold domain (HF) is shown. **b** 3D structure of Dpb4–Dpb3 and Dpb4–Dls1 heterodimers. The histone fold domain of Dpb4 is in orange, while that of Dpb3 and Dls1 is in light blue and green, respectively. A62 is shown as a red stick. The 3D structure of Dpb4–Dpb3 was extracted from PDB 6WJV[35], while the 3D model of Dls1 was built using I-TASSER[90]. **c** HO expression was induced at time zero by galactose addition to G2-arrested cells that were kept arrested in G2 by nocodazole. Relative fold enrichment of Dpb4-HA and Dpb4-A62S-HA compared to untagged Dpb4 (no tag) was evaluated after ChIP with anti-HA antibodies and qPCR analysis. The mean values of three independent experiments are represented with error bars denoting s.d. ***$p < 0.005$ (unpaired two-tailed Student's $t$-test). **d** Western blot with anti-HA antibodies of protein extracts from G2-arrested cells. The experiment was performed independently three times with similar results. **e, f** EMSA with a 61 bp dsDNA and increasing concentrations of Dpb3–Dpb4 and Dpb3–Dpb4[A62S] (**e**), and Dls1–Dpb4 and Dls1–Dpb4[A62S] (**f**) complexes. Bands corresponding to free DNA (F), and protein–DNA complexes with higher stoichiometry (asterisk) are denoted. The experiments were performed independently two times with similar results. **g** Model for Dpb4 function at DSBs. After DSB formation, the Dls1–Dpb4 dimer promotes the association of Isw2 to DSBs, which catalyzes nucleosome sliding/removal and facilitates MRX association to them. The Dpb3–Dpb4 dimer, possibly in complex with Pol ε, in turn, uses its histone chaperone activity to induce re-deposition/exchange of H3 and H4 histones (dark violet), whose subsequent H3 methylation by Dot1 (red dots) leads to H3 exposure to Rad9 recognition.

strain was crossed to *tel1Δ* cells to verify by tetrad dissection that the increased CPT sensitivity cosegregated with *TEL1* deletion and the *KANMX* allele.

**DSB resection**. YEPR exponentially growing cell cultures of JKM139 derivative strains, carrying the HO cut site at the *MAT* locus, were transferred to YEPRG at time zero. *Ssp*I-digested genomic DNA was run on alkaline agarose gels and visualized after hybridization with a single-stranded RNA probe that anneals with the unresected strand at one side of the HO-induced DSB[87]. This probe was obtained by in vitro transcription using Promega Riboprobe System-T7 and plasmid pML514 as a template. Plasmid pML514 was constructed by inserting in the pGEM7Zf *Eco*RI site a 900-bp fragment containing part of the *MAT* locus (coordinates 200870 to 201587 on chromosome III). Quantitative analysis of DSB resection was performed by calculating the ratio of band intensities for ssDNA and the total amount of DSB products. The resection efficiency was normalized with respect to the HO cleavage efficiency for each time point. Densitometric analysis of band intensities was performed using Scion Image Beta 4.0.2 software.

**Spot assays**. Cells grown overnight were diluted to $1 \times 10^7$ cells/mL. 10-fold serial dilutions were spotted on YEPD with or without DNA damaging drugs. Plates were incubated for 3 days at 28 °C.

**Western blotting**. Protein extracts for western blot analysis were prepared by trichloroacetic acid (TCA) precipitation. Frozen cell pellets were resuspended in 100 μL 20% TCA. After the addition of acid-washed glass beads, the samples were vortexed for 10 min. The beads were washed with 200 μL of 5% TCA twice, and the extract was collected in a new tube. The crude extract was precipitated by centrifugation at $850 \times g$ for 10 min. TCA was discarded and samples were resuspended in 70 μL 6× Laemmli buffer (60 mM Tris pH 6.8, 2% SDS, 10% glycerol, 100 mM DTT, 0.2% bromophenol blue) and 30 μL 1 M Tris pH 8.0. Prior to loading, samples were boiled at 95 °C and centrifuged at $850 \times g$ for 10 min. The supernatant containing the solubilized proteins was separated on 10% polyacrylamide gels. Rad53 was detected by using anti-Rad53 polyclonal antibodies (ab104232) (1:2000) from Abcam.

HA- or Myc-tagged proteins were detected by using anti-HA (12CA5) (1:2000) or anti-Myc (9E10) (1:1000) antibodies, respectively.

**Chromatin immunoprecipitation and qPCR**. YEPR exponentially growing cell cultures of JKM139 derivative strains, carrying the HO cut site at the *MAT* locus, were transferred to YEPRG at time zero. Crosslinking was done with 1% formaldehyde for 5 min (Mre11), 10 min (Rad9), or 15 min (Dpb4, Dpb4[A62S], Pol2, Isw2, H3, and H2A). The reaction was stopped by adding 0.125 M glycine for 5 min. Immunoprecipitation was performed by incubating samples with Dynabeads Protein G (ThermoFisher Scientific) for 3 h at 4 °C in the presence of 5 μg anti-HA (12CA5) or anti-Myc antibodies (9E10). H2A and H3 histones were immunoprecipitated by using 5 μg anti-H2A (39945, Active Motif) and 4 μg anti-H3 (ab1791, Abcam) antibodies, respectively. Quantification of immunoprecipitated DNA was achieved by qPCR on a Bio-Rad CFX Connect™ Real-Time System apparatus and Bio-Rad CFX Maestro 1.1 software. Triplicate samples in 20 μL reaction mixture containing 10 ng of template DNA, 300 nM for each primer, 2× SsoFast™ EvaGreen® supermix (Bio-Rad #1725201) (2× reaction buffer with dNTPs, Sso7d-fusion polymerase, MgCl₂, EvaGreen dye, and stabilizers) were run in white 96-well PCR plates Multiplate™ (Bio-Rad #MLL9651). The qPCR program was as follows: step 1, 98 °C for 2 min; step 2, 90 °C for 5 s; step 3, 60 °C for 15 s; step 4, return to step 2 and repeat 40 times. At the end of the cycling program, a melting program (from 65 to 95 °C with a 0.5 °C increment every 5 s) was run to test the specificity of each qPCR. Data are expressed as fold enrichment at the HO-induced DSB over that at the non-cleaved *ARO1* locus, after normalization of the ChIP signals to the corresponding input for each time point. Fold enrichment was then normalized to the efficiency of DSB induction. For histone loss, the fold enrichment from each sample after HO induction was divided by the fold enrichment from uninduced cells, and log₂ of the resulting values was calculated. Oligonucleotides used for qPCR analyses are listed in Supplementary Table 3.

**Coimmunoprecipitation**. Total protein extracts were prepared by breaking cells in 400 μL of buffer containing 50 mM HEPES pH 7.5, 300 mM NaCl, 20% glycerol, 1 mM sodium orthovanadate, 60 mM β-glycerophosphate and protease inhibitor cocktail (Roche Diagnostics). An equal volume of breaking buffer was added to clarified protein extracts and tubes were incubated for 2 h at 4 °C with 50 μL of Protein G-Dynabeads and 5 μg anti-HA (12CA5) antibodies. The resins were then washed twice with 1 mL of breaking buffer and once with 1 mL PBS. Bound proteins were visualized by western blotting with anti-HA (12CA5) (1:2000) or anti-Myc (9E10) (1:1000) antibodies after electrophoresis on a 10 or 15% SDS-polyacrylamide gel.

**Purification of Dpb3–Dpb4 and Dls1–Dpb4 heterodimers**. Design, expression, and purification of Dpb4 heterodimers were performed as previously described[88]. Briefly, the genes coding for Dpb4, Dpb4[A62S], Dpb3, and Dls1 were optimized for the expression in *E. coli* cells and chemically synthesized (Genscript, Piscataway,

NJ, USA). *DPB4* and *DPB4-A62S* genes were cloned in frame with a C-terminal 6xHis-Tag into pET-21a vector (EMD, Millipore, Billerica, MA, USA) between *Nde*I and *Xho*I sites. The *DPB3* and *DLS1* genes were cloned in frame with a C-terminal Strep-Tag into a modified pET-28 vector between *Nco*I and *Xho*I sites. *E. coli* BL21 (DE3) cells were co-transformed with the above plasmids to obtain Dpb3–Dpb4, Dpb3–Dpb4[A62S], Dls1–Dpb4, and Dls1–Dpb4[A62S] heterodimers. Transformed cells were selected on LB agar plates supplemented with ampicillin (100 mg/L) and kanamycin (50 mg/L). Heterodimers were produced in auto-induction ZYM-5052 medium[89] supplemented with ampicillin (100 mg/L) and kanamycin (50 mg/L), extracted and purified by immobilized ion metal affinity chromatography (Jena Bioscience, Jena, Germany) followed by Strep purification on Strep-Tactin resin (IBA Lifesciences, Gottingen, Germany). High-concentrated fractions were buffer-exchanged with phosphate buffer (10 mM, pH 7) by gel filtration on PD-10 columns (GE Healthcare, Little Chalfont, UK). Protein concentration was determined by the Bradford assay (Bio-Rad, Hercules, USA), using bovine serum albumin as a standard. SDS-PAGE was performed on 14% acrylamide gels and stained with Gel-Code Blue (Pierce, Rockford, USA) after electrophoresis. Broad-range, pre-stained molecular-mass markers (GeneSpin, Milan, Italy) were used as standards.

**Circular dichroism spectroscopy**. CD spectra of purified proteins were obtained in phosphate buffer at the concentration of 2 μM with a J-815 spectropolarimeter (JASCO, Europe, Lecco, Italy), using a 0.1-cm path length cuvette. Spectra were collected in the 190-260 range with 0.2 nm data pitch and 20 nm/min scanning speed. All spectra were corrected for buffer contribution, averaged from four independent acquisitions, and smoothed by using a third-order least-square polynomial fit. Thermal denaturation ramps were obtained measuring the variation of CD signal at 208 nm when progressively heating the sample from 25 to 90 °C. Data were analyzed with OriginPro 2020 (OriginLab Corporation, Northampton, USA). Measurements were performed in triplicate.

**Electrophoretic mobility shift assay (EMSA)**. EMSA was performed by incubating 1.5 pmol of 61 bp ³²P-labeled dsDNA (5′-GACGCTGCCGAATTCTAC CAGTGCCTTGCTAGGACATCTTTGCCCACCTGCAGGTTCACCC-3′)[43] with purified Dpb3–Dpb4 and Dpb3–Dpb4[A62S] (0, 0.75, 1.5, 3, 7, 13, 20 pmol) or Dls1–Dpb4 and Dls1–Dpb4[A62S] (0, 20, 40, 80, 100, 120, 160 pmol) in ice for 10 min in binding buffer (20 mm HEPES-NaOH pH 7.5, 0.5 mM EDTA, 0.05% NP-40, 10% (v/v) glycerol and 60 μg/mL BSA) to a final volume of 50 μL. Reactions were loaded on a non-denaturing 6% acrylamide/bisacrylamide gel and separated by running for 2 h at 150 V at 4 °C using a low-ionic strength buffer (6.73 mM Tris-HCl pH 7.5, 3.3 mM NaOAc pH 5 and 1 mM EDTA). Gels were soaked for 15 min in 10% methanol, 10% acetic acid solution, vacuum-dried and exposed to an autoradiography film.

**3D modeling**. The 3D structure of the Dpb3–Dpb4 heterodimer was extracted from PDB 6WJV[35]. The 3D model of Dls1 was predicted by I-TASSER web server[90] and superimposed on the 3D structure of Dpb3 in the Dpb4–Dpb3 heterodimer using Pymol 2.4.1 software. The figures were prepared using UCSF Chimera X 0.93 software[91].

**Statistical analysis**. Statistical analysis was performed using Microsoft Excel Professional 365 software. *P*-values were determined by using an unpaired two-tailed *t*-test. No statistical methods or criteria were used to estimate the size or to include or exclude samples.

**Reporting summary**. Further information on research design is available in the Nature Research Reporting Summary linked to this article.

## Data availability
All data are in the paper and Supplementary Information. The structure of Dpb4/Dpb3 was extracted from PDB 6WJV. All data are available from the authors upon reasonable request. Source data are provided with this paper.

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

## Acknowledgements

We thank J. Haber, J. Diffley, and B. Pfander for providing yeast strains. We also thank G. Lucchini for the critical reading of the manuscript, and M. Villa, F. Esposito, and S. Calabrese for preliminary data. This work was supported by Fondazione AIRC under IG 2017 - ID. 19783 and Progetti di Ricerca di Interesse Nazionale (PRIN) 2017 to M.P.L. E. C. was supported by the Italian Ministry of University and Research (MIUR) through the grant "Dipartimenti di Eccellenza 2017" to the University of Milano Bicocca.

## Author contributions

M.P.L. and M.C. conceived the idea. E.C. performed drop tests in Figs. 1, 4, and 7e, f; DSB resection and the relative analysis in Figs. 2a, b, 6c, d, 7a, Supplementary Fig. 4; ChIP analysis and western blot in Figs. 2c–e, 3a–d, 5, 6a, b, e, f, 7d, g, h, 8c, d, and Supplementary Fig. 3; EMSA assay in Fig. 8e, f. Coimmunoprecipitation in Supplementary Fig. 5. M.G. performed drop tests in Supplementary Fig. 2; western blot analysis in Fig. 7b, c; contributed to perform EMSA in Fig. 8e, f; E.G. performed the screen and identified the *dpb4-A62S* allele. M.M. expressed and purified Dpb3-Dpb4 and Dls1-Dpb4 heterodimers in Supplementary Fig. 6; constructed Fig. 8a, b. M.P.L. and M. C. supervised and coordinated the work. M.P.L wrote the paper. M.P.L., E.C., M.G., and M.C. revised the text.

## Competing interests

The authors declare no competing interests.
