## [Peer Review File · Nature Communications]

REVIEWER COMMENTS

Reviewer #1 (Remarks to the Author):

Dpb4 is a highly conserved histone-fold protein. Dpb4 is known to interact with two distinct chromatin complexes, the DNA polymerase ϵ (Pol ϵ) holoenzyme and the ISW2/hCHRAC chromatin remodeling complex. Casari et al. in this manuscript showed that Dpb4 is involved in two aspects of sensing and processing DNA double-strand breaks (DSBs): 1) Dpb4 promotes the DSB resection by interacting with the ISW2 complex to remove histones from the DSB ends; 2) Dpb4 mediates checkpoint function by interacting Dpb3 to promote Rad9 association to DSBs. They further showed that the Dpb4 checkpoint function is enhanced by the A62S mutation in Dpb4, and it acts in the same pathway of Dot1, the histone H3 K79 methyltransferase, in promoting Rad9 association to DSBs. The results are of general interest as they established a physical and functional link between Dpb4 and the DSB repair. The experiments were well designed and carefully executed, and most of the results are convincing. There are several concerns as listed below.

Major concerns:

1. It is still unclear how Dpb4-A62S enhances Rad53 activation while the lack of Dpb4 reduces it. The study could be strengthened by examining how the A62S mutation affects the interaction of Dpb4 with Dpb3, Dls1 and DNA.
2. It is unclear whether just the Dpb3-Dpb4 heterodimer or the Pol ϵ holoenzyme is recruited to DSB sites. ChIP analysis of Pol2 or Dpb2 at DSB sites would address this question.
3. It is important to demonstrate that the tagged proteins used are functional, such as Dpb4-HA, Dpb4-A62S-HA, and Rad9-HA.

Minor concern:

1. Is the A62S mutation localized within the histone fold domain or other region of Dpb4? It would be helpful to provide a figure indicating the location of the mutation.

Reviewer #2 (Remarks to the Author):

The authors presented evidence of a dual role of Dpb4 in the repair of DSB. By association with the ISW2 complex, Dpb4 promotes nucleosome eviction from DSB facilitating the recruitment of MRX and subsequent DSB resection. Furthermore, in conjunction with Dpb3, Dpb4 induces the activation of the DNA damage checkpoint by promoting Rad9 association with DSB sites.

The authors show how the nucleosome eviction function of Dpb4 is related to the ISW2 complex activity. However, it's not clear if the Rad9-activating function of Dpb3-Dpb4 depends on the whole Pol ϵ holoenzyme. As the authors mention in the discussion, Pol2 is involved in DNA replication checkpoint activation and this function is independent from the DNA damage checkpoint activation. This hints at a standalone function of Dpb3-Dpb4 in Rad9 activation but there is no evidence to prove it. This article would benefit with further insight into this issue.

To address this, the authors could test the association of Pol2 or Dpb2 at DSB in WT and *dpb4-A62S* cells. The increased persistence of Dpb4-A62S leads to increased Rad9 recruitment. Therefore, if the Dpb4 checkpoint function depends on the Pol ϵ holoenzyme, the *dpb4-A62S* mutation should also result in an increased presence of Pol2/Dpb2 at DSB.

The authors also speculate that Dpb3-Dpb4 could stimulate redeposition/exchange of histones H3-H4 at DSB ends, which could facilitate the methylation of H3K79 by Dot1. However, the authors don't consider a simpler possibility where Dpb3-Dpb4 could be directly recruiting Rad9 to DSB sites. To test this possibility, they could test if there is a physical interaction between Dpb3-Dpb4 and Rad9.

Other issues:

Fig. 2B. it is not clear how many times the experiments were performed. Statistical analysis of Fig. 2B should be presented.

Fig. 2E: Statistically analysis among all strains and time points should be presented.

Fig. 3E. it is not clear whether the increase of Dpb4-A62S-HA is due to an increase of the proteins at DNA replication forks. This should be tested using synchronized cells.

Fig. 6D. Statistical analysis should be presented.

Reviewer #3 (Remarks to the Author):

This manuscript characterises a mutation in the histone fold protein Dpb4 that exacerbates the DNA damage sensitivity of TEL1 null mutants. The mutant was initially identified in a screen that has been previously reported. Interestingly the mutation (A62S) enhances the DNA binding of Dpb4. Previously it has been shown that Dpb4 is part of two separate dimeric histone fold complexes, Dpb3-Dpb4, which is associated with the Pol epsilon holoenzyme, and Dis1-Dpb4, which acts within the ISW2 complex. Interestingly, the data separate out two functions affected by Dpb4-A62S: first it compromises the ability of ISW2 to promote efficient resection at DSBs (assayed by a standard HO-dependent resection assay) and second it promotes the Dot1- and H3-K79 methylation-dependent association of Rad9 around the DSB site, resulting in enhanced checkpoint signalling. The data produced are fully consistent with the interpretation that A62A increases the stability of DNA binding, resulting in compromised ISW2 activity and enhanced Rad9 recruitment. The increased affinity of Dpb4-A62S with DNA is not experimentally tested.

The manuscript is, in general, very clearly presented and the data interpreted appropriately. The quality of the experiments are generally excellent. The work will be of interest to scientists studying the DSB repair and pathway choice. I have the following comments:

1. It would help the reader if a supplementary figure was included that diagrammatically explains the resection assay, including an explanation for the uniform intensity of r1 and why the relative intensity rather than total intensity of the bands is the key metric. Size markers should be indicated on figure 2A, 6C and S1.
2. The section "Dpb4 promotes histone removal near DSBs" is a little confusing to read and needs a little more explanation to be convincing. For example "This process is thought to hamper...". "This process" appears to refer to histone sliding, which is thought to enhance resection? and "As resection of the DSB ends leads to...". I assume this refers to a potential 50% reduction in the event of full resection?
3. A bit of discussion as to why gamma-H2A is protective (i.e. the triple mutants being more sensitive). The one additional experiment that should be performed is to establish that Rad9 is still recruited and in the H2A mutant background.
4. The main paragraph on page 13 is confusingly written and should be restructured. Particularly, the relevance of the 1st sentence to the Dot1 work is not clear. I would recommend introducing Dot1 and H3 methylation first. It is intriguing that the recruitment of Rad9 in the dbp4-A62S mutant appears to be independent of gamma H2A. Establishing this experimentally would be reassuring (see point 3) and would allow this paragraph to be less convoluted.
5. Why is the data for dpb3 null presented as supplementary and not as part of figure 6? This is a really key point to the manuscript.

Response to reviewers' comments

Reviewer #1

Dpb4 is a highly conserved histone-fold protein. Dpb4 is known to interact with two distinct chromatin complexes, the DNA polymerase ϵ (Pol ϵ) holoenzyme and the ISW2/hCHRAC chromatin remodeling complex. Casari et al. in this manuscript showed that Dpb4 is involved in two aspects of sensing and processing DNA double-strand breaks (DSBs): 1) Dpb4 promotes the DSB resection by interacting with the ISW2 complex to remove histones from the DSB ends; 2) Dpb4 mediates checkpoint function by interacting Dpb3 to promote Rad9 association to DSBs. They further showed that the Dpb4 checkpoint function is enhanced by the A62S mutation in Dpb4, and it acts in the same pathway of Dot1, the histone H3 K79 methyltransferase, in promoting Rad9 association to DSBs. The results are of general interest as they established a physical and functional link between Dpb4 and the DSB repair. The experiments were well designed and carefully executed, and most of the results are convincing. There are several concerns as listed below.

Major concerns:

1. It is still unclear how Dpb4-A62S enhances Rad53 activation while the lack of Dpb4 reduces it. The study could be strengthened by examining how the A62S mutation affects the interaction of Dpb4 with Dpb3, Dls1 and DNA.

We have examined whether the A62S mutation affects the interaction between Dpb4 and Dpb3 and/or Dls1 by coimmunoprecipitation (new Supplementary Fig. 5) and we found that the A62S mutation does not seem to affect Dpb3-Dpb4 and Dls1-Dpb4 complex formation.

Furthermore, to test whether the A62S mutation might affect the interaction of Dpb4 with DNA, we have expressed and purified Dpb3-Dpb4, Dpb3-Dpb4^{A62S}, Dls1-Dpb4 and Dls1-Dpb4^{A62S} heterodimers from *E. coli* cells. When we tested them for the ability to bind DNA in a gel electrophoretic mobility shift assay, we found that wild type Dpb3-Dpb4 and Dls1-Dpb4 complexes were capable to shift the dsDNA oligomer into a distinct slower migrating band (new Fig. 8e, f). Both Dpb3-Dpb4^{A62S} and Dls1-Dpb4^{A62S} generated a similar slower migrating band although less efficiently compared to the corresponding wild type complexes (new Fig. 8e, f). However, they both showed the appearance of a second band (new Fig. 8e, f), suggesting that the A62S amino acid substitution favors transition to higher order Dpb3-Dpb4-DNA and Dls1-Dpb4-DNA complexes that can explain the increased amount of Dpb4 bound to DSBs detected by CHIP.

2. It is unclear whether just the Dpb3-Dpb4 heterodimer or the Pol ϵ holoenzyme is recruited to DSB sites. CHIP analysis of Pol2 or Dpb2 at DSB sites would address this question.

We measured Pol2 association at the HO-induced DSB in *dpb4* Δ and *dpb4-A62S* cells. To exclude possible effects of DNA replication, HO expression was induced by galactose addition to G2-arrested cells that were kept arrested in G2 with nocodazole. Following HO induction by galactose addition, Pol2 was recruited to the HO-induced DSB (new Fig. 7g) and the A62S mutation increased Pol2 occupancy at the HO-induced DSB, whereas the lack of Dpb4 decreased it (new Fig. 7g), suggesting that Dpb4 acts through Pol ϵ to promote checkpoint activation in response to DSBs.

3. It is important to demonstrate that the tagged proteins used are functional, such as Dpb4-HA, Dpb4-A62S-HA, and Rad9-HA.

We now show that cells expressing Mre11-Myc, Dpb4-HA and Rad9-HA tagged proteins were not sensitive to DNA damaging agents compared, respectively, to *mre11* Δ , *dpb4* Δ

and *rad9* Δ cells (new Supplementary Fig. 2). Furthermore, cells expressing Rad9-STAA-HA and Dpb4-A62S-HA were as sensitive to DNA damaging agents as cells expressing untagged Rad9-STAA and Dpb4-A62S (Supplementary Fig. 2).

Minor concern:

1. Is the A62S mutation localized within the histone fold domain or other region of Dpb4? It would be helpful to provide a figure indicating the location of the mutation.

Sequence and structural analyses indicate that the A62 residue is localized on the α 2 helix within the histone fold domain and interacts with the residues I74 and I87 on the α 3 helix of the histone fold domain of Dpb3 and Dls1, respectively. The location of the mutation is now shown in the new Fig. 8a and b.

Reviewer #2

The authors presented evidence of a dual role of Dpb4 in the repair of DSB. By association with the ISW2 complex, Dpb4 promotes nucleosome eviction from DSB facilitating the recruitment of MRX and subsequent DSB resection. Furthermore, in conjunction with Dpb3, Dpb4 induces the activation of the DNA damage checkpoint by promoting Rad9 association with DSB sites.

The authors show how the nucleosome eviction function of Dpb4 is related to the ISW2 complex activity. However, it's not clear if the Rad9-activating function of Dpb3-Dpb4 depends on the whole Pol ϵ holoenzyme. As the authors mention in the discussion, Pol2 is involved in DNA replication checkpoint activation and this function is independent from the DNA damage checkpoint activation. This hints at a standalone function of Dpb3-Dpb4 in Rad9 activation but there is no evidence to prove it. This article would benefit with further insight into this issue.

To address this, the authors could test the association of Pol2 or Dpb2 at DSB in WT and *dpb4*-A62S cells. The increased persistence of Dpb4-A62S leads to increased Rad9 recruitment. Therefore, if the Dpb4 checkpoint function depends on the Pol ϵ holoenzyme, the *dpb4*-A62S mutation should also result in an increased presence of Pol2/Dpb2 at DSB. We measured Pol2 association at the HO-induced DSB in *dpb4* Δ and *dpb4*-A62S cells. To exclude possible effects of DNA replication, HO expression was induced by galactose addition to G2-arrested cells that were kept arrested in G2 with nocodazole. Following HO induction by galactose addition, Pol2 was recruited to the HO-induced DSB (new Fig. 7g) and the A62S mutation increased Pol2 occupancy at the HO-induced DSB, whereas the lack of Dpb4 decreased it (new Fig. 7g), suggesting that Dpb4 acts through Pol ϵ to promote checkpoint activation in response to DSBs.

The authors also speculate that Dpb3-Dpb4 could stimulate redeposition/exchange of histones H3-H4 at DSB ends, which could facilitate the methylation of H3K79 by Dot1. However, the authors don't consider a simpler possibility where Dpb3-Dpb4 could be directly recruiting Rad9 to DSB sites. To test this possibility, they could test if there is a physical interaction between Dpb3-Dpb4 and Rad9.

We agree with the reviewer that the simpler hypothesis was that Dpb3-Dpb4 complex directly recruit Rad9 at DSBs. However, we have no evidence supporting this hypothesis because we failed to detect interaction between Dpb4 and Rad9 by coimmunoprecipitation. We now discuss this hypothesis in the discussion section.

Other issues:

Fig. 2B. it is not clear how many times the experiments were performed. Statistical analysis of Fig. 2B should be presented.

We now reported in the Figure legend that the experiments were performed three times. Since it is difficult to insert the p-values in the graphs, the statistical analysis is shown in the Source data file and this is mentioned in the figure legend.

Fig. 2E: Statistically analysis among all strains and time points should be presented.

Since it is difficult to insert the p-values in the graph, the statistical analysis is shown in the Source data file and this is mentioned in the figure legend.

Fig. 3E. it is not clear whether the increase of Dpb4-A62S-HA is due to an increase of the proteins at DNA replication forks. This should be tested using synchronized cells.

To exclude that the increased Dpb4-A62S association at DSBs can be due to an increase of proteins at the replication forks, we have repeated the experiments in G2-arrested cells and we found very similar results (new Fig. 8c).

Fig. 6D. Statistical analysis should be presented.

Since it is difficult to insert the p-values in the graph, the statistical analysis is shown in the Source data file and this is mentioned in the figure legend.

Reviewer #3

This manuscript characterises a mutation in the histone fold protein Dpb4 that exacerbates the DNA damage sensitivity of TEL1 null mutants. The mutant was initially identified in a screen that has been previously reported. Interestingly the mutation (A62S) enhances the DNA binding of Dpb4. Previously it has been shown that Dpb4 is part of two separate dimeric histone fold complexes, Dpb3-Dpb4, which is associated with the Pol epsilon holoenzyme, and Dis1-Dpb4, which acts within the ISW2 complex. Interestingly, the data separate out two functions affected by Dpb4-A62S: first it compromises the ability of ISW2 to promote efficient resection at DSBs (assayed by a standard HO-dependent resection assay) and second it promotes the Dot1- and H3-K79 methylation-dependent association of Rad9 around the DSB site, resulting in enhanced checkpoint signalling. The data produced are fully consistent with the interpretation that A62A increases the stability of DNA binding, resulting in compromised ISW2 activity and enhanced Rad9 recruitment. The increased affinity of Dpb4-A62S with DNA is not experimentally tested.

See response to reviewer 1 point 1, regarding the affinity of Dpb4-A62S for DNA.

The manuscript is, in general, very clearly presented and the data interpreted appropriately. The quality of the experiments are generally excellent. The work will be of interest to scientists studying the DSB repair and pathway choice. I have the following comments:

1. It would help the reader if a supplementary figure was included that diagrammatically explains the resection assay, including an explanation for the uniform intensity of r1 and why the relative intensity rather than total intensity of the bands is the key metric. Size markers should be indicated on figure 2A, 6C and S1.

The resection assay is now shown in Supplementary Fig. 1. Size markers are now indicated in the resection gels (new Fig. 2a, 6c and 7a). We mentioned in the text that, as previously observed (Shroff et al., 2004), all the resection products are quite uniform and persisted throughout the experiment, suggesting that resection initiates asynchronously

after DSB formation. Furthermore, we measured the relative intensity of each resection band to normalize them against the total cut DNA fragment.

2. The section "Dpb4 promotes histone removal near DSBs" is a little confusing to read and needs a little more explanation to be convincing. For example "This process is thought to hamper...". "This process" appears to refer to histone sliding, which is thought to enhance resection? and "As resection of the DSB ends leads to...". I assume this refers to a potential 50% reduction in the event of full resection?

We have modified this section and mentioned in the text that resection should lead to a 50% reduction at a maximum.

3. A bit of discussion as to why gamma-H2A is protective (i.e. the triple mutants being more sensitive). The one additional experiment that should be performed is to establish that Rad9 is still recruited und in the H2A mutant background.

See response to point 4.

4. The main paragraph on page 13 is confusingly written and should be restructured. Particularly, the relevance of the 1st sentence to the Dot1 work is not clear. I would recommend introducing Dot1 and H3 methylation first. It is intriguing that the recruitment of Rad9 in the *dbp4*-A62S mutant appears to independent of gamma H2A. Establishing this experimentally would be reassuring (see point 3) and would allow this paragraph to be less convoluted.

We have modified and reorganized the entire paragraph (see also point 3). To make it less convoluted, we have substituted panels a and b of Figure 5 with ChIP assays showing Rad9 association at the HO-induced DSB in *hta1*-S129A and *dpb4Δ hta1*-S129A (new Fig. 5c). We found that Rad9 association at DSBs was decreased in *dpb4Δ hta1*-S129A double mutant compared to each single mutant, indicating that Dpb4 function in promoting Rad9 association at DSBs occurs independently of Rad9-γH2A interaction.

5. Why is the data for *dpb3* null presented as supplementary and not as part of figure 6? This is a really key point to the manuscript.

The resection assay for *dpb3Δ* is now presented in the new Fig. 7a.

REVIEWERS' COMMENTS

Reviewer #1 (Remarks to the Author):

The authors addressed all my concerns; the manuscript is significantly improved.

Reviewer #2 (Remarks to the Author):

My concerns were largely addressed and i support the acceptance of the present study.

Reviewer #3 (Remarks to the Author):

The authors have added significant new data and have satisfied my (minor) concerns very well.

Response to reviewers' comments

We thank all the reviewers for the positive comments.

Reviewer #1:

The authors addressed all my concerns; the manuscript is significantly improved.

Reviewer #2:

My concerns were largely addressed and I support the acceptance of the present study.

Reviewer #3:

The authors have added significant new data and have satisfied my (minor) concerns very well.